# Sensitivity Analysis of the Frequency Response Function of Carbon-Fiber-Reinforced Plastic Specimens for Different Direction of Carbon Fiber as Well as Spectral Loading Pattern

**DOI:** 10.3390/ma12182983

**Published:** 2019-09-15

**Authors:** Chan-Jung Kim

**Affiliations:** Department of Mechanical Design Engineering, Pukyong National University, 45 Yongso-ro, Nam-gu 48513, Korea; cjkim@pknu.ac.kr; Tel.: +82-51-629-6169

**Keywords:** scaled sensitivity index, spectral loading pattern, direction of carbon fiber, resonance frequency, damping coefficient

## Abstract

Carbon-fiber-reinforced plastic (CFRP) has been used in many industries owing to its excellent specific-strength characteristics; however, the control of its mechanical properties is difficult owing to the directivity nature of carbon fiber as well as the composition of layered structures. In addition, the damping coefficient of CFRP varies with spectral loading patterns under random and harmonic excitation owing to the high values of damping characteristics compared with conventional steel materials. A scaled sensitivity index was proposed to compare the magnitude of the frequency response function over two parameters of interest: the direction of the carbon fiber and the spectral loading pattern for CFRP specimens. Three specimens with different directions (0°, 45°, and 90°) were prepared and uniaxial excitation testing was conducted for two different spectral loading cases: random and harmonic. The summation of the frequency response was used to calculate the sensitivity index to eliminate the effects of the location of measurement data, and all sensitivity indexes were calculated using the measured responses. Finally, the sensitivity of each CFRP specimen was discussed for two cases, i.e., the direction of carbon fiber and the spectral loading pattern, using the scaled sensitivity index results.

## 1. Introduction

Carbon-fiber-reinforced plastic (CFRP) is a next-generation lightweight material with outstanding specific-strength characteristics that can replace steel and aluminum materials, and basic research and product development are underway [1,2,3,4,5,6,7,8]. The mechanical properties of CFRPs are determined by the conditions of the carbon fiber and polymer resin constituting the product. In addition, the combination of textile materials to form a carbon fiber structure greatly affects the mechanical properties of the product. Because many candidate materials, such as plain weave, twill weave, and unidirectional ones exist as major materials for general use in CFRPs, diverse layered structures can be formed by combining these [9,10,11,12,13]. The mechanical properties and test results differ from the layered structure when CFRP is applied to products. In particular, the direction of the carbon fiber plays an important role in the determination of the mechanical properties of the CFRP structure; hence, the directivity of the CFRP material should be designed according to the principle direction of external forces.

Damping reduces the external energy of a load by converting it into internal energy using the inherent characteristics of a material; this provides great advantages in terms of durability by enhancing structural stability and diminishing response [14,15,16]. The damping coefficient is used in the time domain or frequency domain to represent damping. The accurate identification of a system's damping coefficient plays a critical role in understanding the mechanical characteristics of the system. Various methodologies for identification exist. In particular, frequency damping values corresponding to each resonance point are required to represent damping in the frequency domain, which are referred to as modal damping coefficients. The values are usually identified by modal testing using devices such as an impact hammer, and CFRPs have been reported with particularly large damping values compared with other steel materials [17,18,19]. Combined with its excellent specific strength, CFRPS have received much attention for application in mechanical products. Previous studies verified that the dynamic response changes according to the spectral loading patterns for conventional steel material in the uniaxial excitation test [20,21]. If CFRPs exhibit different damping characteristics under different excitation profile conditions, then differences in absolute values could inevitably cause errors larger than those of other materials.

Sensitivity analysis is applied to mechanical systems to identify the variations in response or system parameters with small changes in design variables, in order to derive an effective solution to achieve satisfactory system characteristics with minimum design changes. Sensitivity analysis may be classified into three types: finite difference, analytic, or semi-analytic methods [22,23,24,25,26]. Recently, the design index can also be calculated from the response data without knowledge of the dynamics of the target system [27,28]. In any case, it is possible to identify variations in the characteristics of the target system with slight changes in conditions or design modifications so that it becomes an effective method for comparing CFRP specimens according to the parameters of interest, direction of the carbon fiber, or spectral loading patterns.

Sensitivity analysis equations were derived from the frequency response functions for the CFRP material, and a scaled sensitivity index was proposed to compare all CFRP specimens according to the parameters of interest, direction of the carbon fiber, and spectral loading patterns. The proposed theoretical formulation was verified using a uniaxial excitation test for three CFRP specimens with different directions of the carbon fiber (0°, 45°, and 90°). Two spectral loading cases were considered under random and harmonic conditions. The uniaxial excitation test was performed with sine-sweep excitation as well as a random excitation, and the frequency response function was acquired using the excitation load and acceleration sensors at seven different positions. The scaled sensitivity analysis was conducted using the measured frequency response functions. The sensitivity index for all specimens were compared in terms of the two parameters of interest, i.e., the direction of the carbon fiber and the damping ratio. The dynamics of CFRP specimens was discussed from the sensitivity results.

## 2. Theoretical Background

Assuming that the system of interest is linear, the frequency response function is used to obtain stiffness or damping values in the frequency domain, where excitation is applied and the response is received at the desired point. The frequency response function is one of the most efficient methods of representing the dynamic characteristics of an object in the linear domain. Excitation is applied with an impact hammer or exciter, for example, and the displacement, velocity, or acceleration can serve as the response. As the damping values for the CFRP used in this study are reported to be higher than those of existing steel materials, and that the material may exhibit relatively strong nonlinear characteristics (which is a common characteristic of composite materials), the exciter test method was selected instead of the impact type. This approach has the advantage of noncontact testing. The excitation load can be precisely controlled, and an averaged frequency response function with high reliability can be obtained by repeated testing in the exciter-based model test.

The governing equation of the linear mechanical system is expressed as the simple one-degree-of-freedom (1-DOF) Equation (1), with the basic mechanical components of mass (*m*), damping coefficient (*c*), and stiffness coefficient (*k*), and external force *f*(*t*). The damped natural frequency of the mechanical system can be obtained by the simple calculation of scalar parameters in Equation (2), and the damping ratio can be obtained by Equation (3).
(1)$$m\ddot{x}+c\dot{x}+kx=F$$
(2)$$\omega_{d}=\omega_{n}\sqrt{\left(1-\xi^{2}\right)},\;\omega_{n}=\sqrt{\frac{k}{m}}$$
(3)$$\xi =\frac{c}{2\sqrt{mk}}$$

The simple system in Equation (1) can be transformed into a modal coordinate by mass normalization in Equation (4). The general N-DOF system can be expressed with modal decomposition, as shown in Equations (5)–(7) [14,15].
(4)$$\ddot{x}+2\omega_{n}\xi \dot{x}+\omega_{n}^{2}x=f,\;f\left(t\right)=\frac{F}{m}$$
(5)$$\left[\begin{array}{ccc} 1 & & zeros\\ & \ddots & \\ zeros & & 1 \end{array}\right]\ddot{X}+\left[\begin{array}{ccc} 2\omega_{n,1}\xi_{1} & & zeros\\ & \ddots & \\ zeros & & 2\omega_{n,N}\xi_{N} \end{array}\right]\dot{X}+\left[\begin{array}{ccc} \omega_{1}^{2} & & zeros\\ & \ddots & \\ zeros & & \omega_{N}^{2} \end{array}\right]X=\left[\begin{array}{c} F_{1}\\ \vdots \\ F_{N} \end{array}\right]$$
where, $\omega_{n,i}$, $\xi_{i}$, and $F_{i}$ are the natural frequency, damping ratio, and external force in the i-th mode, respectively, and $X={\left[\begin{array}{ccc} x_{1} & \ldots & x_{N} \end{array}\right]}^{T}$ is the column vector in the modal coordinate. If the linear system is assumed to be a single input condition, that is $F_{i}=0$ except for i≠j, then the frequency response function over the input $F_{j}$ can be expressed as Equation (6).
(6)$$\frac{R\left(\omega \right)}{F_{j}\left(\omega \right)}=H\left(\omega \right)=\overset{N}{\underset{i=1}{\sum }}\frac{R_{i}^{e}}{-\omega^{2}+2\omega_{n,i}\omega \xi_{i}j+\omega_{n,i}^{2}}$$

The linearly time-invariant (LTI) system has unique system variables, such as mass, damping coefficient, and stiffness coefficient. The frequency response function in Equation (6) is considered as one of the unique characteristics of the LTI system [14,15]. However, the unique dynamics of the LTI system cannot be preserved if one of the system variables are changed by the external force condition. In this study, any nonlinear property stemming from the external force was not considered. The frequency response function H(ω) could be a function of both system parameters $\omega_{n,i}$ and $\xi_{i}$ in the case where the variation of residue $R_{i}^{e}$ is small enough to be negligible under an external force $F_{j}\left(\omega \right)$.

The strength of the CFRP specimen has directional characteristics according to the direction of the carbon fiber and changes the stiffness coefficient of the specimen. Since the mass is a constant value for the same material and volume shape, the natural frequency could be changed according to the variation of the coefficient of stiffness from Equation (2). Thus, the natural frequency $\omega_{n,i}$ should be expressed by $\omega_{n,i}\left(\theta \right)$, according to the different directions of the carbon fiber. On the other hand, a previous study showed that the coefficient of damping changes according to the spectral loading of external force for a simple specimen with S45C [20,21]. It is also well recognized that the damping coefficient of CFRP materials is much higher than that of conventional steel materials [17,18,19]. Thus, if the spectral loading pattern is denoted as p, it is possible to express the damping coefficient of the CFRP specimen as $\xi_{i}\left(p\right)$ instead of the constant value $\xi_{i}$ in Equation (6). In addition, the direction of the carbon fiber may affect the damping coefficient of the CFRP specimen because the directivity of the CFRP material changes all the structural properties, including the structural strength. Therefore, the final expression of the damping coefficient $\xi_{i}\left(p\right)$ is replaced with $\xi_{i}\left(p,\theta \right)$, which includes the influence of the directivity of the carbon fiber.

Two parameters $\omega_{n,i}\left(\theta \right)$ and $\xi_{i}\left(p,\theta \right)$ can be obtained from the measured frequency response function between the input force and any response location on the CFRP specimen. The damped natural frequency in Equation (2) is measured from the *i*th peak point of the measured frequency response function. When two frequencies around the damped natural frequency in which the energy value is halved are defined as $\omega_{d,i}^{\left(1\right)}$ and $\omega_{d,i}^{\left(2\right)}$, the following relationship holds [14,15].
(7)$$\xi_{i}\left(p,\theta \right)=\frac{\left|\omega_{d,i}^{\left(2\right)}-\omega_{d,i}^{\left(1\right)}\right|}{2\omega_{d,i}}\;$$

Then, the natural frequency $\omega_{n,i}\left(\theta \right)$ can be derived from the damped natural frequency $\omega_{d,i}$ from Equation (2). Considering these relationships between the two parameters *p* and θ, the frequency response function of the CFRP specimen should be rewritten as
(8)$$H\left(\omega ,\theta ,p\right)=\overset{N}{\underset{i=1}{\sum }}\frac{R_{i}^{e}}{-\omega^{2}+2\omega_{n,i}\left(\theta \right)\omega \xi_{i}\left(\theta ,p\right)j+\omega_{n,i}{\left(\theta \right)}^{2}}\;$$

The partial derivatives of the frequency response function can be calculated for the two parameters of interest as follows.
(9)$$\frac{\partial \left(H\left(\omega ,\theta ,p\right)\right)}{\partial \theta }=\overset{N}{\underset{i=1}{\sum }}\frac{-2R_{i}^{e}}{{\left[-\omega^{2}+2\omega_{n,i}\left(\theta \right)\omega \xi_{i}\left(\theta ,p\right)j+\omega_{n,i}{\left(\theta \right)}^{2}\right]}^{2}}\left\{\left(\omega \xi_{i}\left(\theta ,p\right)j+\omega_{n,i}\left(\theta \right)\right)\frac{\partial \left(\omega_{n,i}\left(\theta \right)\right)}{\partial \theta }\;+\omega \omega_{n,i}\left(\theta \right)j\frac{\partial \left(\xi_{i}\left(\theta ,p\right)\right)}{\partial \theta }\right\}$$
(10)$$\frac{\partial \left(H\left(\omega ,\theta ,p\right)\right)}{\partial p}=\overset{N}{\underset{i=1}{\sum }}\frac{-2R_{i}^{e}}{{\left[-\omega^{2}+2\omega_{n,i}\left(\theta \right)\omega \xi_{i}\left(\theta ,p\right)j+\omega_{n,i}{\left(\theta \right)}^{2}\right]}^{2}}\left\{\omega \omega_{n,i}\left(\theta \right)j\frac{\partial \left(\xi_{i}\left(\theta ,p\right)\right)}{\partial p}\right\}\;$$

The derivatives in Equations (9) and (10) have the frequency response function itself, and the magnitude of each derivative could be directly related to the sensitivity result [22,23,24,25,26,27,28]. Hence, for the sensitivity index for each parameter of interest in the *i*-th mode, specimen #k is formulated as the magnitude of the partial derivative after dividing it by the magnitude of the frequency response function, as shown in Equations (11) and (12), respectively.
(11)$$I_{\theta ,i}\left(k\right)=\left|\frac{0.5}{H\left(\omega ,\theta ,p\right)}\frac{\partial \left(H\left(\omega ,\theta ,p\right)\right)}{\partial \theta }\right|\;=\left|\omega_{n,i}\left(\theta \right)\frac{\partial \left(\omega_{n,i}\left(\theta \right)\right)}{\partial \theta }+\omega \left(\xi_{i}\left(\theta ,p\right)\frac{\partial \left(\omega_{n,i}\left(\theta \right)\right)}{\partial \theta }+\omega_{n,i}\left(\theta \right)\frac{\partial \left(\xi_{i}\left(\theta ,p\right)\right)}{\partial \theta }\right)j\right|$$
(12)$$I_{p,i}\left(k\right)=\left|\frac{0.5}{H\left(\omega ,\theta ,p\right)}\frac{\partial \left(H\left(\omega ,\theta ,p\right)\right)}{\partial p}\right|=\left|\omega \omega_{n,i}\left(\theta \right)\frac{\partial \left(\xi_{i}\left(\theta ,p\right)\right)}{\partial p}\right|\;$$

Finally, the scaled sensitivity index in the *i*-th mode can be formulated for specimen #*k* as follows.
(13)$${\tilde{I}}_{\theta ,i}=\frac{I_{\theta ,i}\left(k\right)}{\sum_{k=1}^{3}I_{\theta ,i}\left(k\right)}\;$$
(14)$${\tilde{I}}_{p,i}=\frac{I_{p,i}\left(k\right)}{\sum_{k=1}^{3}I_{p,i}\left(k\right)}$$

## 3. Uniaxial Excitation Test

Because CFRP specimens show different characteristics according to the direction of the carbon fiber, a vibration test was conducted for specimens with different directions to measure the respective damping coefficients, as well as the resonance frequencies. CFRP specimens were prepared from a large CFRP mother plate by cutting them in three different directions to remove any errors caused by the properties of the CFRP material. The pre-impregnated composite fibers (T700 carbon fiber, Toray, Japan) were manufactured by SK Chemicals, South Korea, and stacked onto a 12-layer unidirectional CFRP mother plate with 3 mm thickness. The three directions of carbon fibers in the CFRP specimens were designated as 0°, 45°, and 90°, as shown in Figure 1. The specimens were referred to as specimen #1, specimen #2, and specimen #3, respectively. Although the CFRP specimens were prepared from the same mother CFRP plate, different dynamic behaviors of the specimens could be observed depending on the direction of the carbon fiber under uniaxial excitation.

The frequency response function can be obtained through both the input force from the exciter and the responses of specimens. It was possible to calculate the damping coefficient and damped natural frequencies within the frequencies of interest. The force information, as well as response accelerations, were measured simultaneously during the excitation test. To realize the conditions using an exciter, one lengthwise end of the CFRP specimen was fixed tightly with a jig and excited vertically with an exciter attached to the fixed location.

Although the testing environments for obtaining the frequency response function were identical, two methods were used to apply the load with the exciter. A single input frequency was increased when applying the vibration in the harmonic method, while all frequencies were applied simultaneously in random excitation. Table 1 and Table 2 list the excitation profiles used in the test. Since the frequency response function is only valid under the time-invariant linear system, the nonlinear properties of the CFRP specimens were eliminated from the averaged spectral data of more than 100 frequency response data in the same vibration test mode.

During the vibration excitation test under the above conditions, the input excitation data was collected from a load sensor attached beneath the excited jig. The response accelerations were collected at different locations from the acceleration sensors attached to the CFRP specimens. The input load data and response data of the acceleration sensor were all converted to the frequency domain, and were then converted in the frequency domain to obtain the frequency response function. The measurement sensors are shown in Figure 2. Because the acceleration values differ by location, the acceleration was measured and data was acquired from sensors at diverse locations.

The jigs were assembled by fastening the upper and lower jigs tightly with two bolts after preparing two rectangular jigs to attach at one end of the CFRP, as shown in Figure 3. SUS304 was used for the rectangular jig material because it was stiff enough to clamp the specimen safely. The jig must be designed as a structure with high stiffness exhibiting rigid-body behavior with an excitation frequency from 10 Hz to 500 Hz.

## 4. Sensitivity Analysis

The frequency response function for the three specimens (#1~#3) was obtained from the measured data between the force input and responses of the specimens; however, the averaging method of the frequency response function differed according to the spectral loading pattern. In the case of random excitation, the frequency response function was calculated for all measurement ranges, and the values for each frequency were averaged to obtain the values for a single averaged frequency response function. Because of the characteristics of random excitation, completely simultaneous excitation in all frequencies was not possible; hence, linear averaging for all frequency response functions should be conducted to eliminate noise factors from each spectral line. In contrast, because the input data values changed sequentially under harmonic excitation, the largest value for each frequency changed according to the sequence of single spectral inputs from 10 Hz to 500 Hz. Thus, the measured frequency response functions should be averaged by the peak-hold method to preserve any effective responses under the single-input condition.

Another problem in the selection of the frequency response function is the location of the response on the CFRP specimen. The spectral acceleration at the specimen will respond differently depending on the location because each measurement location has a unique magnitude and phase according to the mode shape of the CFRP specimen. This means that the frequency response function at a certain location is limited to providing information about the local response. Therefore, the summation of the frequency response function was considered to represent the unique global frequency response function from seven measurement positions (#1–#7). The global frequency response function of the CFRP specimens was calculated by the summation of each frequency element from all measured frequency responses from #1 to #7. The summation of the frequency response function was calculated for specimens #1–#3, as shown in Figure 4, Figure 5 and Figure 6, respectively.

Using the measured frequency response functions, the modal analysis of CFRP specimens was conducted and the normalized mode shape of each specimen was plotted in Figure 7 and Figure 8, respectively. It was verified that the first mode corresponds to the bending mode, regardless of the direction of the carbon fiber; the bending shape was also similar among all specimens. The second mode was difficult to define because the mode shape changed with the direction of the carbon fiber; different shapes of twisting mode for 0° and 45°, and bending mode for 90°.

The damped natural frequency and damping coefficient were calculated from the summation of frequency response functions in Figure 4, Figure 5 and Figure 6, respectively; the calculation results are summarized in Table 3. Here, the error between two spectral loading cases is defined as below, and $r_{K}$ denotes the modal parameter at K excitation mode (R: random, H: harmonic).
(15)$$Error\left(\%\right)=\frac{\left|r_{R}-r_{H}\right|}{r_{H}}$$

The frequency error between two spectral loading cases was lower than those from the damping error situation in Table 3. This means that the sensitivity of the damping coefficient is higher than that of resonance frequency over the spectral loading patterns. In particular, the resonance frequency error was less than 0.7% and 2.1% in the first and second mode, respectively. Thus, the assumption used in Equation (8), i.e., the partial derivative of resonance frequency over the spectral loading pattern is zero, was verified by the experimental results.

The resonance frequency over the direction of the carbon fiber decreased gradually in the first mode because the structural rigidity of the CFRP specimen decreased under the uniaxial testing condition, with the clamped rectangular specimen illustrated in Figure 1. The mode shapes in all specimens showed similar bending modes in Figure 7. In the second mode, the resonance frequency increased slowly as the directivityof the carbon fiber increased. However, the mode shapes in all specimens showed different behaviors: a certain twisting mode in specimen #1, a different twisting mode in specimen #2, and bending mode in specimen #3. Thus, comparison of the resonance frequencies of the three specimens may be less valuable except in the second resonance mode, and it is reasonable to conclude that the mode shape of the CFRP specimen is very sensitive to the direction of the carbon fiber. The variations in resonance frequency showed a similar trend in both spectral loading patterns, and the statement holds in both cases.

The damping coefficients increased gradually in the first and second modes, but the results do not coincide with the resonance frequency cases. In the first mode, the starting value of the damping coefficient was high under harmonic excitation. The rate of increase of the damping coefficient was also high under random excitation such that the final damping coefficient at the 90° direction of the carbon fiber was higher under random excitation compared with that under harmonic excitation. This means that the sensitivity of the damping coefficient to changes in spectral loading patterns should be considered in the sensitivity analysis, and shows good agreement with the theoretical frequency response function model formulated in Equation (8). In the second mode, the rate of increase of the damping coefficient was high under random excitation compared with the harmonic case. The starting value of the damping coefficient at 0° direction of the carbon fiber remained high under harmonic excitation.

The variations in modal parameters, resonance frequency, and damping coefficient according to the direction of the carbon fiber are plotted in Figure 9 and Figure 10. The curve-fitted second-order polynomial lines were also plotted to calculate the partial derivative for each direction of the carbon fiber. The constants of the curved-fitted line and the partial derivative values according to the direction of the carbon fiber are summarized in Table 4.

Using the partial derivative results in Table 4, the scaled sensitivity index of the direction of the carbon fiber can be obtained using Equation (13). In the case of scaled sensitivity according to the spectral loading pattern (see Equation (14)), the partial derivative of the damping coefficient over the spectral loading pattern ($=\partial \left(\xi_{i}\left(\theta ,p\right)\right)/\partial p$) is required to calculate it. Since only two spectral patterns were used in this study, the partial derivative over the spectral loading pattern was applied by the relative error summarized in Table 3. Since the sensitivity index denotes the sensitivity of the frequency response function over small variations in the parameter of interest, resonance frequency, or damping ratio, there is no need to match the partial derivative results in Table 4 and the sensitivity results of the frequency response function. In addition, the direction of the carbon fiber can be mapped onto the CFRP specimens, which means that specimens #1, #2, and #3 are directly related to 0°, 45°, and 90°, respectively. Considering these conditions, the scaled sensitivity index can be compared for four conditions for all specimens, as illustrated in Figure 11, Figure 12, Figure 13 and Figure 14.

The sensitivity index over the direction of the carbon fiber is very high at specimen #1 and decreased rapidly as the direction of the carbon fiber increased in the first mode; the sensitivity results show the opposite tendency in the second mode. The sensitivity order changed over the spectral loading patterns in the first mode, but the highest value with specimen #1 did not change at all. This means that it is reasonable to use specimen #1 under the current uniaxial excitation condition owing to the highest structural rigidity, but the frequency response function could also be the most sensitive over the small variations in the direction of the carbon fiber. On the other hand, specimen #3 became the most sensitive specimen in the second mode; this result may be the opposite of that of the first mode. However, comparison of the sensitive index in the second mode has less valuable physical meaning because the mode shapes differ from each other over the direction of the carbon fiber, as seen in Figure 8. The sensitivity index over the spectral loading pattern is also very high at specimen #1, while the other specimens have lower sensitivity values in the first mode. This means that the frequency response function of specimen #1 will change easily over small variations in spectral loading patterns, and the effect of the dynamics of external forces is an important issue for the specimen with 0° direction of the carbon fiber. In case of the second mode, the sensitivity index is similar for all specimens, as seen in Figure 14. Thus, the variation of the frequency response function against the spectral loading pattern will be similar for all specimens in the second mode.

Specimen #1 has the highest structural stiffness as a result of assigning the carbon fiber in the principle direction (0°) to support the rectangular specimen against uniaxial excitation. However, both sensitivity indexes indicate that specimen #1 is the most sensitive CFRP structure over the direction of the carbon fiber as well as the spectral loading pattern in the first bending mode. Hence, it is very important to minimize differences in the direction of the carbon fiber in all CFRP structures. Different dynamic responses are also expected from variations in the spectral loading patterns. In the second mode, the direction of the carbon fiber affects the mode shape of the CFRP specimen, even if the variation in resonance frequency did not change significantly. The sensitivity index for the direction of carbon fiber indicates the highest value in specimen #3, but direct comparison was not a significant issue owing to the difference in the mode shape of each specimen.

## 5. Results

Specimen #1 with 0° direction of the carbon fiber was the most sensitive to small variations in the direction of carbon fiber as well as spectral loading patterns in the first mode. Hence, quality control inspections should be conducted to minimize errors in the direction of carbon fiber in CFRP products. In addition, the response of specimen #1 has the disadvantage of changing its magnitude with spectral loading patterns compared with other specimens. In the second mode, the dynamics of each specimen changed rapidly according to the direction of the carbon fiber. The mode shape of each specimen differed from each other, although the resonance frequency did not significantly change such that the sensitivity index derived in the second mode has less physical meaning. These analysis results indicate that the proposed scaled sensitivity index is an efficient method to evaluate CFRP specimens according to the direction of the carbon fiber. The sensitivity result may serve as a useful guideline for engineers in designing CFRP materials.

## 6. Conclusions

The frequency response function for CFRP specimens was derived by considering the nature of system variables over two parameters: the direction of the carbon fiber and the damping ratio. A scaled sensitivity index was then proposed for each parameter of interest. Partial derivatives of the frequency response function were obtained from the curve-fitted line of the measured response data during uniaxial excitation testing, and the corresponding scaled sensitivity index was calculated for specimens #1 to #3. The mode shapes of CFRP specimens were shown to have similar bending modes in the first mode, but showed different shapes in the second mode. Specimen #1 is the most promising candidate for uniaxial excitation in the first mode because its structural rigidity is maximized with 0° direction of the carbon fiber for uniaxial excitation.

## Figures and Tables

**Figure 1 materials-12-02983-f001:**
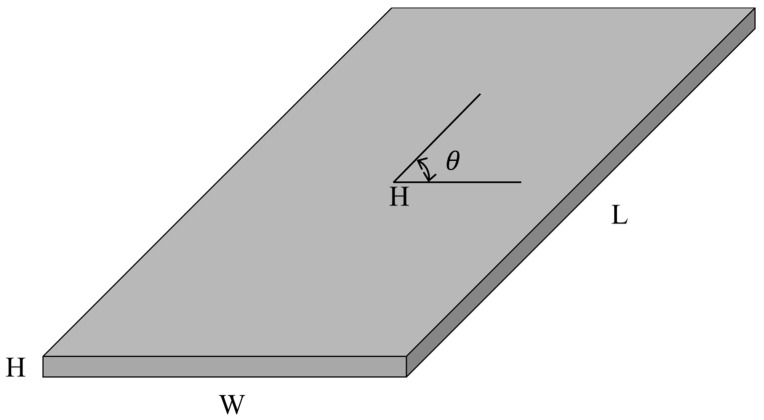
Configuration of a specimen of carbon-fiber-reinforced plastic (CFRP) material: L: 150 mm, W: 80 mm, H: 3 mm, *θ* = 0°, 45°, 90°.

**Figure 2 materials-12-02983-f002:**
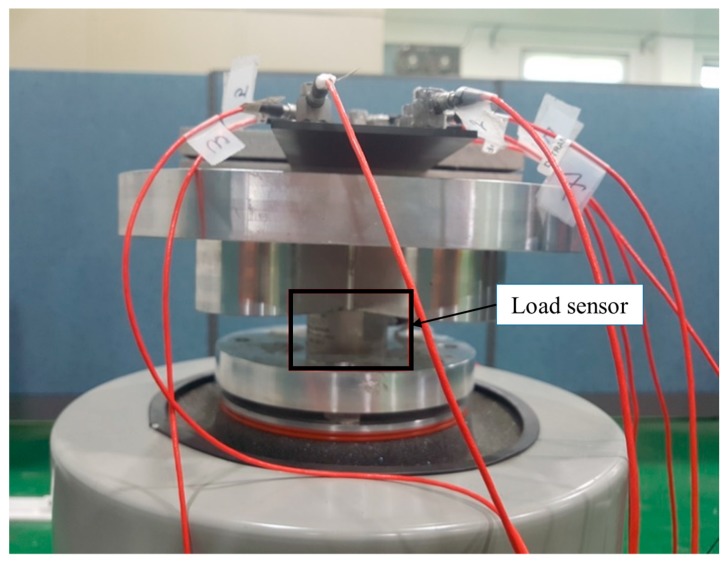
Load sensor on excitation jig.

**Figure 3 materials-12-02983-f003:**
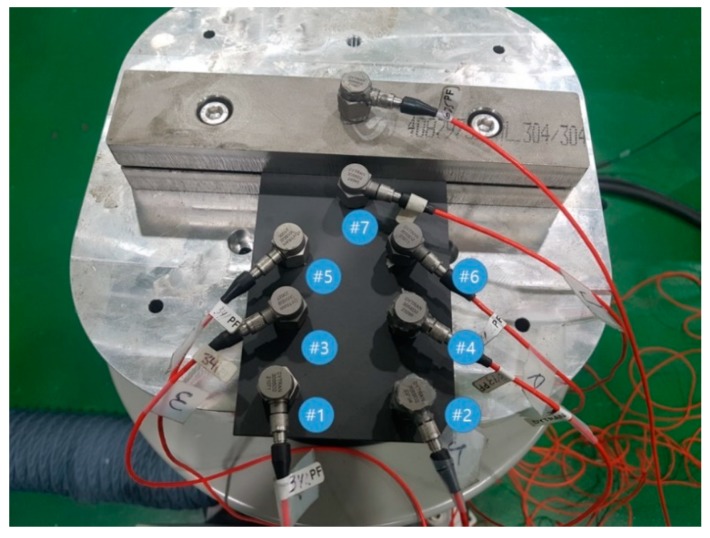
Configuration of uniaxial vibration: the clamping location and sensor locations (#1~#7).

**Figure 4 materials-12-02983-f004:**
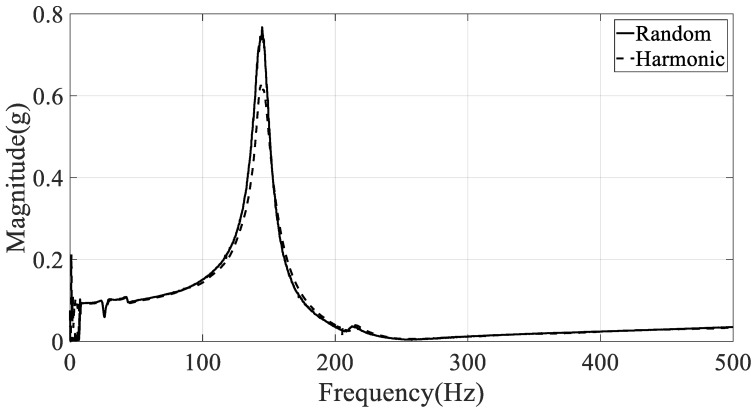
Summation of frequency response functions with CFRP specimen #1 (direction: 0°)

**Figure 5 materials-12-02983-f005:**
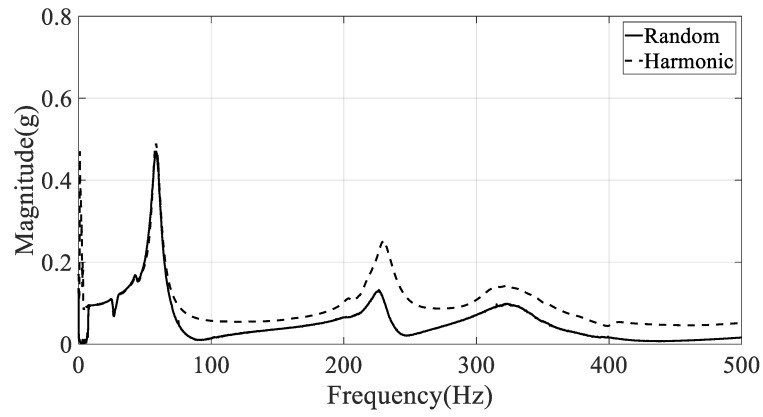
Summation of frequency response functions with CFRP specimen #2 (direction: 45°).

**Figure 6 materials-12-02983-f006:**
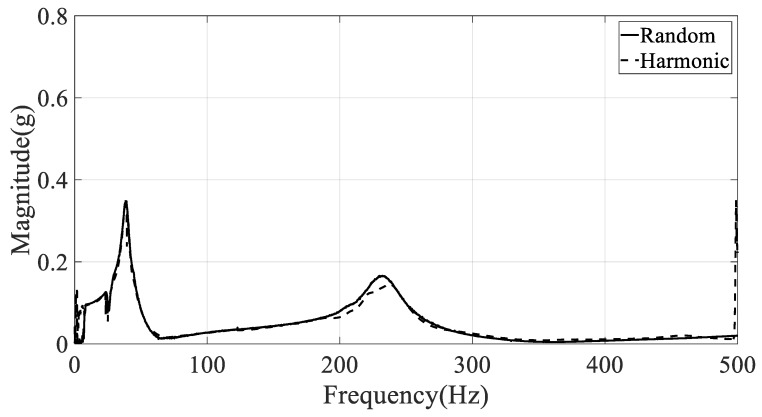
Summation of frequency response functions with CFRP specimen #3 (direction: 90°).

**Figure 7 materials-12-02983-f007:**
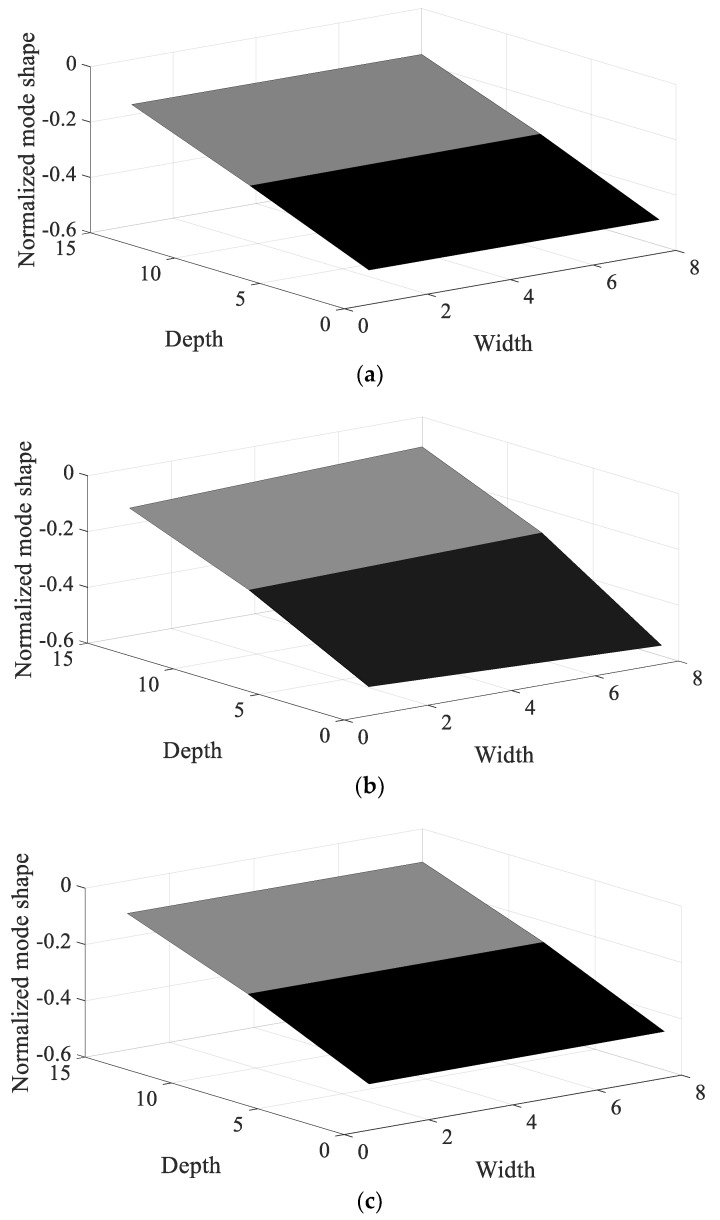
Normalized mode shape of first mode. (**a**) Specimen #1, (**b**) Specimen #2, (**c**) Specimen #3.

**Figure 8 materials-12-02983-f008:**
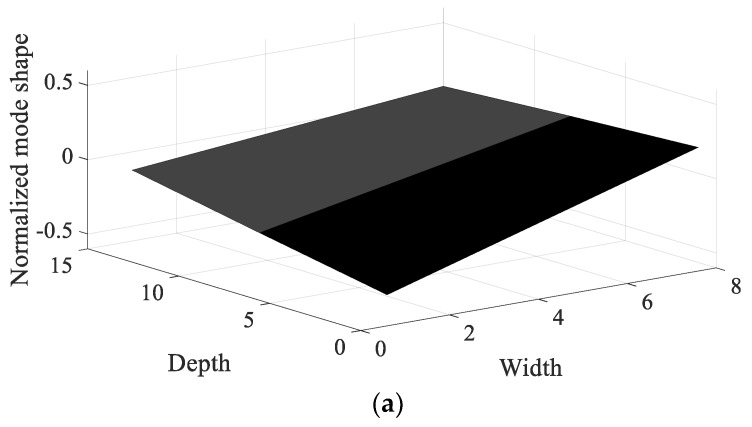
Normalized mode shape of second mode. (**a**) Specimen #1, (**b**) Specimen #2, (**c**) Specimen #3.

**Figure 9 materials-12-02983-f009:**
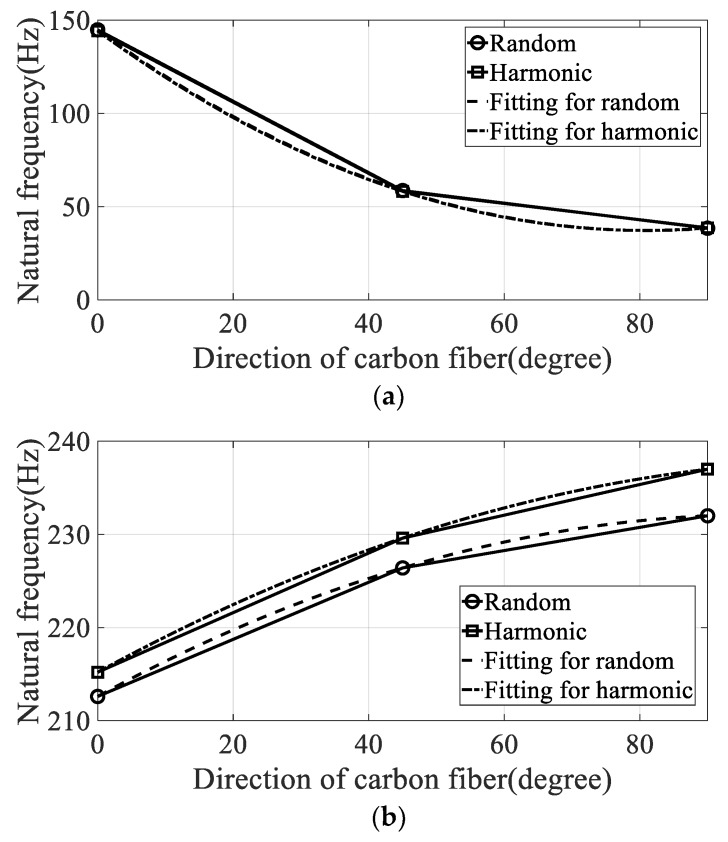
Variation of natural frequency over direction of carbon fiber. (**a**) First mode, (**b**) second mode.

**Figure 10 materials-12-02983-f010:**
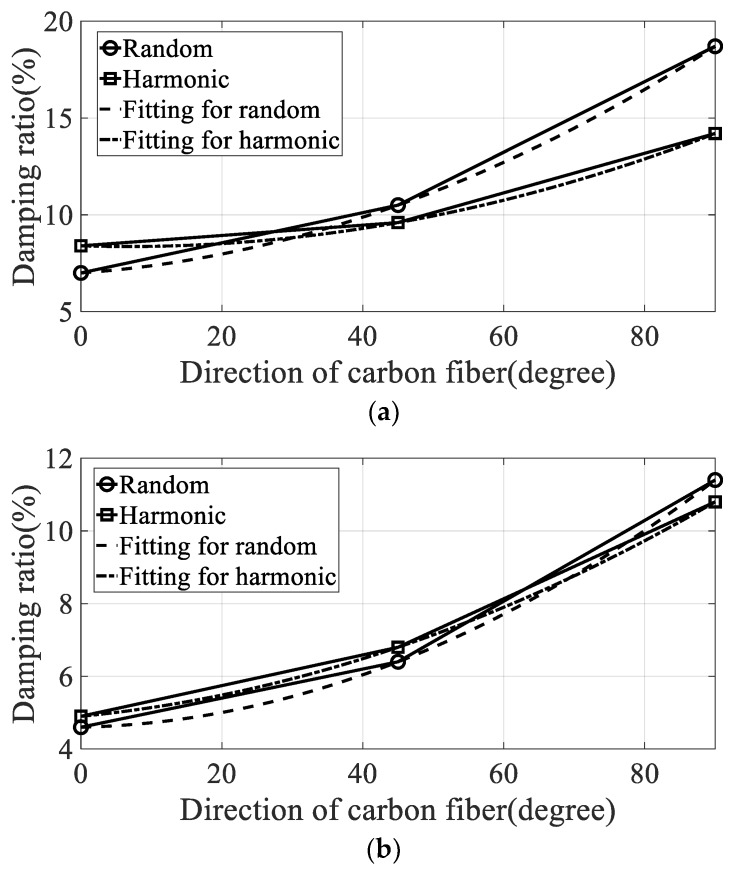
Variation of damping ratio over direction of carbon fiber. (**a**) First mode, (**b**) second mode.

**Figure 11 materials-12-02983-f011:**
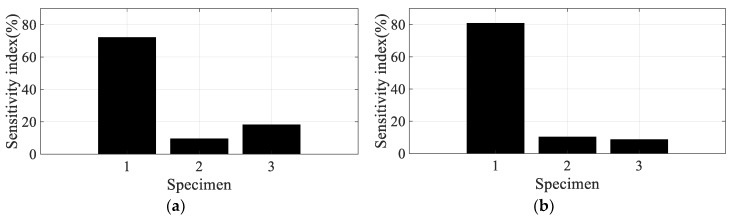
Scaled sensitivity index over the direction of carbon fiber in the first mode. (**a**) Random (**b**) Harmonic.

**Figure 12 materials-12-02983-f012:**
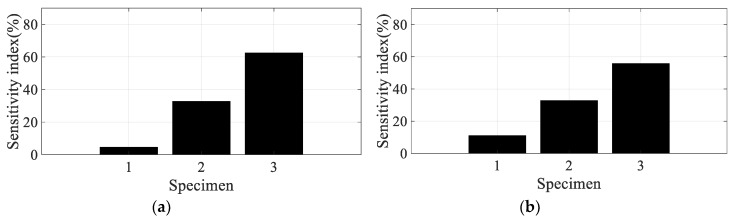
Scaled sensitivity index over the direction of carbon fiber in the second mode. (**a**) Random (**b**) Harmonic.

**Figure 13 materials-12-02983-f013:**
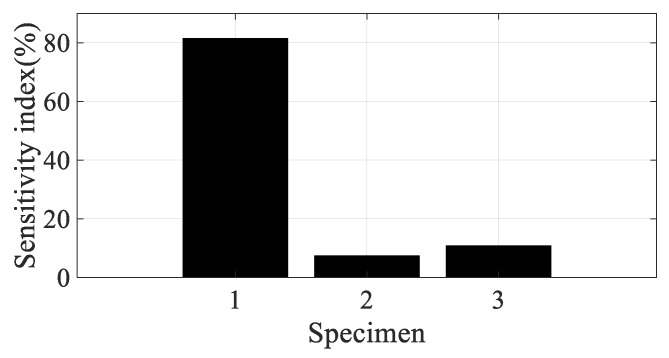
Scaled sensitivity index over the spectral loading pattern in the first mode.

**Figure 14 materials-12-02983-f014:**
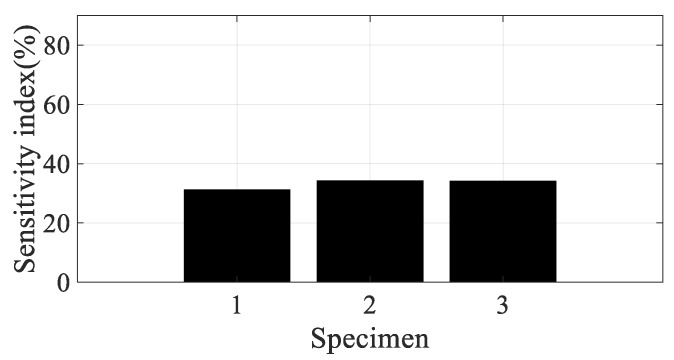
Scaled sensitivity index of damping ratio in the second mode.

**Table 1 materials-12-02983-t001:** Profile for random excitation.

No.	Frequency (Hz)	Acceleration (g^2^/Hz)
1	10	0.005
2	500	0.005

**Table 2 materials-12-02983-t002:** Profile for harmonic excitation.

No.	Frequency (Hz)	Acceleration (g)
1	10	0.5
2	500	0.5

**Table 3 materials-12-02983-t003:** Summation of experimental modal parameters of CFRP specimens.

Direction	Random Excitation		Harmonic Excitation		Frequency Error (%)	Damping Error (%)
	Resonance (Hz)	Damping Coefficient (*ξ*)	Resonance (Hz)	Damping Coefficient (*ξ*)		
0 (Specimen I)	144.8	7.0%	144.2	8.4%	−0.4	16.7
	212.6	4.6%	215.2	4.9%	1.2	6.1
45 (Specimen II)	58.6	10.5%	58.2	9.6%	−0.7	−9.4
	226.4	6.4%	229.6	6.8%	1.4	5.9
90 (Specimen III)	38.4	18.7%	38.6	14.2%	0.5	−31.7
	232.0	11.4%	237.0	10.8%	2.1	−5.6

**Table 4 materials-12-02983-t004:** Constants in curved fitted lines and derivative results according to the direction of carbon fiber.

Item	Mode	Fitted Line Function		Partial Derivative					
		Random (R)	Harmonic (H)	θ=0(Degree)		θ=45(Degree)		θ=90(Degree)	
				R	H	R	H	R	H
Natural Freq.	#1	$0.016\theta^{2}-2.65\theta +144.8$	$0.016\theta^{2}-2.65\theta +144.8$	−2.65	−2.65	−1.18	−1.17	0.28	0.30
	#2	$-0.002\theta^{2}+0.40\theta +212.6$	$-0.0017\theta^{2}+0.40\theta +215.2$	0.40	0.40	0.22	0.24	0.033	0.087
Damp. ratio	#1	$0.0012\theta^{2}0.026\theta +7.0$	$0.008\theta^{2}-0.011\theta +8.4$	0.026	−0.01	0.13	0.064	0.23	0.14
	#2	$0.0008\theta^{2}+0.0044\theta +4.6$	$0.0005\theta^{2}+0.019\theta +4.9$	0.0044	0.019	0.076	0.066	0.15	0.11

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
