# Peer review of "Sensitivity Analysis of the Frequency Response Function of Carbon-Fiber-Reinforced Plastic Specimens for Different Direction of Carbon Fiber as Well as Spectral Loading Pattern"

_materials, 2019, doi:10.3390/ma12182983_

Round 1

Reviewer 1 Report

The article is well written and deals with a topic of interest appropriate to the chosen journal. Some revision is necessary to better understand and replicate the experiments:
in chapter 3 it would be necessary to indicate the type of plastic material used and also the relative suppliers, as it is not indicated.

In the last chapter 5 the results are presented in general, I would suggest a division of the last chapter (5) in two different sections: The results and an other chapter dedicated only to conclusions.

Author Response

Author's reply was written in a attached file.

Reviewer 2 Report

- The subject matter of the paper is within the scope of the journal and has a  good technical quality.

- I remarked:

the originality and the clarity of the paper. That the paper represents a new and an original contribution in the field.

- The references are adequate and all they are necessary.

- Author, please arrange in a nicer form the Equations (ex. (11)) and insert point or comma after, as it is required..

- Table 4 is out of the page format.

- I think, the author must strengthen the References section with the references that use the same technique, to make the technique used more plausible, for instance:

Stress singularity of symmetric free-edge joints with elasto-plastic behaviour, Computational Materials Science 52(1), 282-286, (2012); Considerations on mixed initial-boundary value problems for micropolar porous bodies, Syst. Appl 25(1-2), 175-196, (2016); Influence of Fiber Volume Content on Thermal Conductivity in Transverse and Fiber Direction of Carbon Fiber-Reinforced Epoxy Laminates,Materials12(7), 1084, (2019)

If the authors take into account all these recommendations, then the manuscript deserves to be published.

Author Response

(The authors gave the same response as above.)
